# Cost-utility analysis of Coronary Artery Calcium screening to guide statin prescription among intermediate-risk patients in Thailand

Pakpoom Wongyikul[1], Phichayut Phinyo[2]*, Pannipa Suwannasom[3], Apichat Tantraworasin[4], Surasak Saokaew[5,6,7,8]*

1 Department of Biomedical Informatics and Clinical Epidemiology (BioCE), Faculty of Medicine, Chiang Mai University, Chiang Mai, Thailand; Center for Clinical Epidemiology and Clinical Statistics, Faculty of Medicine, Chiang Mai University, Chiang Mai, Thailand, 2 Department of Biomedical Informatics and Clinical Epidemiology (BioCE), Faculty of Medicine, Chiang Mai University, Chiang Mai, Thailand; Center of Multidisciplinary Technology for Advanced Medicine, Faculty of Medicine, Chiang Mai University, Chiang Mai, Thailand, 3 Division of Cardiology, Department of Internal Medicine, Faculty of Medicine, Chiang Mai University, Chiang Mai, Thailand, 4 General Thoracic Unit, Department of Surgery, Faculty of Medicine, Chiang Mai University Hospital, Chiang Mai, Thailand, 5 Division of Social and Administrative Pharmacy, Department of Pharmaceutical Care, School of Pharmaceutical Sciences, University of Phayao, Phayao, Thailand, 6 Unit of Excellence on Clinical Outcomes Research and Integration (UNICORN), School of Pharmaceutical Sciences, University of Phayao, Phayao, Thailand, 7 Center of Health Outcomes Research and Therapeutic Safety (Cohorts), School of Pharmaceutical Sciences, University of Phayao, Phayao, Thailand, 8 Center of Excellence in Bioactive Resources for Innovative Clinical Applications, Chulalongkorn University, Bangkok, Thailand

☯ These authors contributed equally to this work.
* phichayutphinyo@gmail.com (PP); surasak.sa@up.ac.th (SS)

## Abstract

### Objective

This study aims to evaluate the cost-utility of Coronary Artery Calcium (CAC) screening for primary prevention in Thai patients with intermediate cardiovascular disease (CVD) risk, compared to the current practice according to the ACC/AHA 2019 guideline recommendation without the use of a CAC score.

### Methods

A hybrid model combining a decision tree and a Markov model was constructed to compare costs and QALYs from a societal perspective. The model evaluated a target population of statin-naïve individuals aged 40–75 with intermediate CVD risk. We assessed the impact of statin initiation for primary prevention based on the ACC/AHA 2019 guideline with CAC screening compared to without CAC screening over a 35-year time horizon. The service costs and related household expenses were based on the Thai setting. The incremental cost-effectiveness ratio (ICER) was compared against the official willingness-to-pay threshold of Thailand (160,000 THB, approximately 4,400 USD per QALY). Probabilistic and additional one-way sensitivity

which permits unrestricted use, distribution, and reproduction in any medium, provided the original author and source are credited.

**Data availability statement:** All relevant data are within the paper.

**Funding:** This study was partially funded by the University of Phayao and Thailand Science Research and Innovation Fund (Fundamental Fund 2025, Grant No. 5017/2567).The funders had no role in study design, data collection and analysis, decision to publish, or preparation of the manuscript.

**Competing interests:** The authors have declared that no competing interests exist.

analyses were performed to assess the robustness of the model and evaluate how variations in key assumptions impact the results. These analyses help determine the reliability of the findings by exploring the extent to which changes in input parameters influence the overall conclusions.

## Results

The CAC screening strategy required an incremental cost of 10,091 THB to gain 0.62 QALYs per person, resulting in an ICER of 16,308 THB per QALY gained. For the probabilistic sensitivity analysis, at the official Thai threshold, the probability of cost-effectiveness was 71% for CAC screening. Sensitivity analyses based on varying the effect of drug adherence, drug cost, incidence of CVD events, and the distribution of CAC scores demonstrated robust cost-effectiveness favouring CAC screening.

## Conclusion

CAC screening strategy is cost-effective in the Thai context, especially when the cost of screening and high-potency statins is low.

## Introduction

Atherosclerotic cardiovascular diseases (ASCVD), caused by the gradual buildup of cholesterol plaques inside the arteries, are the leading cause of death globally, with ischaemic heart disease and cerebrovascular disease being the two major sources of disability [1,2]. Many studies have demonstrated the heavy economic burden of ASCVD, particularly in low- and middle-income countries (LMICs) compared to high-income countries (HICs) [3,4]. Effective management of dyslipidaemia, a disorder characterised by abnormal levels of lipids in any form, is crucial for lowering the incidence of ASCVD. Nevertheless, numerous LMICs encounter substantial difficulties in executing effective and comprehensive national strategies for the primary prevention of ASCVD [5,6].

In subclinical individuals, primary prevention of ASCVD is based on the predicted 10-year risk of a CVD event. In Thailand, patients are assessed using Thai cardiovascular (CV) risk tools, which estimate their 10-year risk as a percentage [7]. Statin therapy is not recommended for individuals at low risk (<10% 10-year risk), while it is recommended for those at high ASCVD risk (>20% 10-year risk) [8]. However, for individuals at intermediate risk (10% to 20% 10-year risk), treatment is generally only recommended if serum cholesterol levels exceed a defined threshold or if there are other ASCVD risk enhancers present, such as elevated blood pressure, metabolic syndrome, or chronic kidney disease [8–11]. Evidence indicates a substantial heterogeneous baseline risk within the intermediate risk category [12,13]. Shared decision-making between doctor and patient is, therefore, the principle guiding whether a patient in this risk group should initiate statin therapy. However, in Thailand, the treatment initiation rate for dyslipidaemia is notably lower compared to diabetes and hypertension [6].

The Coronary Artery Calcium (CAC) score, introduced in the late 1990s, is a non-invasive imaging technique used to assess coronary artery calcification [14]. It quantifies calcified atherosclerotic plaques within the coronary arteries and has been strongly associated with CVD events [15,16]. Extensive evidence strongly supports the potential of Coronary Artery Calcium (CAC) scores as an accurate tool for cardiovascular risk stratification [17,18]. Previous health economic studies have demonstrated the cost-effectiveness of CAC screening for primary prevention [19–24]. Based on the American College of Cardiology/American Heart Association (ACC/AHA) Cholesterol Management Guideline 2019, CAC screening for informed decisions in intermediate-risk patients appears reasonable [9]. However, the Thailand Clinical Practice Guideline on Pharmacologic Therapy of Dyslipidaemia for Atherosclerotic Cardiovascular Disease (ASCVD) Prevention 2016 did not implement the supplemental use of CAC scores for risk stratification [8]. Additionally, there is a lack of local evidence to support the cost-effectiveness of CAC screening in the Thai context. Consequently, the use of CAC screening in the country is currently guided by the 2019 ACC/AHA recommendations [9]. The inclusion of CAC for ASCVD risk screening in the publicly financed health insurance scheme needs to be justified [25]. This study aims to evaluate the cost-effectiveness of using a CAC screening strategy to guide statin initiation for primary prevention in Thai patients with intermediate ASCVD risk, compared to current practice based on the ACC/AHA 2019 guidelines without incorporating CAC screening.

## Methods

### Model overview

A hybrid model combining a decision tree and a Markov model was constructed (Figs 1 and 2) using Microsoft Excel with the Plant-A-Tree add-in [26]. Quality-adjusted life-years (QALYs) were used as the outcome measure. The model evaluated the target population of statin-naïve individuals aged 40–75 with intermediate cardiovascular (CV) risk, assessed using Thai CV risk tools [9], who were free of known CVD events (myocardial infarction (MI), stroke, or CV death).

This study was conducted in compliance with the process of Thai health technology assessment (HTA) guidelines, which consider transparency, accountability, inclusiveness, timeline, quality, consistency, and contestability for the HTA process. In addition, for economic evaluation, we followed the Thai guidelines and the Consolidated Health Economic Evaluation Reporting Standards (CHEERS) guidelines [27,28]. The study was granted ethical approval from the Institutional Review Board and Ethics Committee of the Faculty of Medicine, Chiang Mai University (HOS 2566-0099).

### Model structure and strategies

A decision tree was used to compare the costs and consequences of two screening strategies: (1) CAC screening and (2) current practice (Fig 1). For the CAC screening strategy, individual patients are assessed for their CAC score to guide statin therapy. A moderate-potency statin is initiated for patients with a CAC score of 1–99, and a high-potency statin is initiated for patients with a CAC score of ≥100. Patients with a CAC score of zero are recommended to reassess the score every 5 years until a positive CAC score is detected [6]. All simulated individuals are assumed to receive lifestyle advice.

The current practice reflects the implementation of the ACC/AHA guideline 2019 without incorporating CAC screening [6], where statin therapy is initiated under the following conditions: for a low-density lipoprotein cholesterol (LDL-C) of 70–189mg/dL, a moderate-potency statin is provided, and for an LDL-C of ≥190mg/dL, a high-potency statin is provided. All patients are assumed to maintain their statin potency level without stepping down. We did not directly compare the CAC screening strategy to the Thai clinical guidelines 2016 [8] because they are likely underutilised due to being outdated and may not reflect current practices.

The Markov model was used to assess natural disease progression in each simulated patient with different baselines. The model tracked quality of life, cost, and time spent in one of the following eight health states: (1) No CVD, (2) Post non-fatal MI, (3) Post recurrent non-fatal MI, (4) Post non-fatal stroke, (5) Post recurrent non-fatal stroke, (6) Post non-fatal MI and stroke, (7) Post recurrent non-fatal MI and stroke, and (8) Dead. Each simulated patient started in the "No CVD" state at age 40 and was followed until age 75 (time horizon: 35 years) (Fig 2) [6].

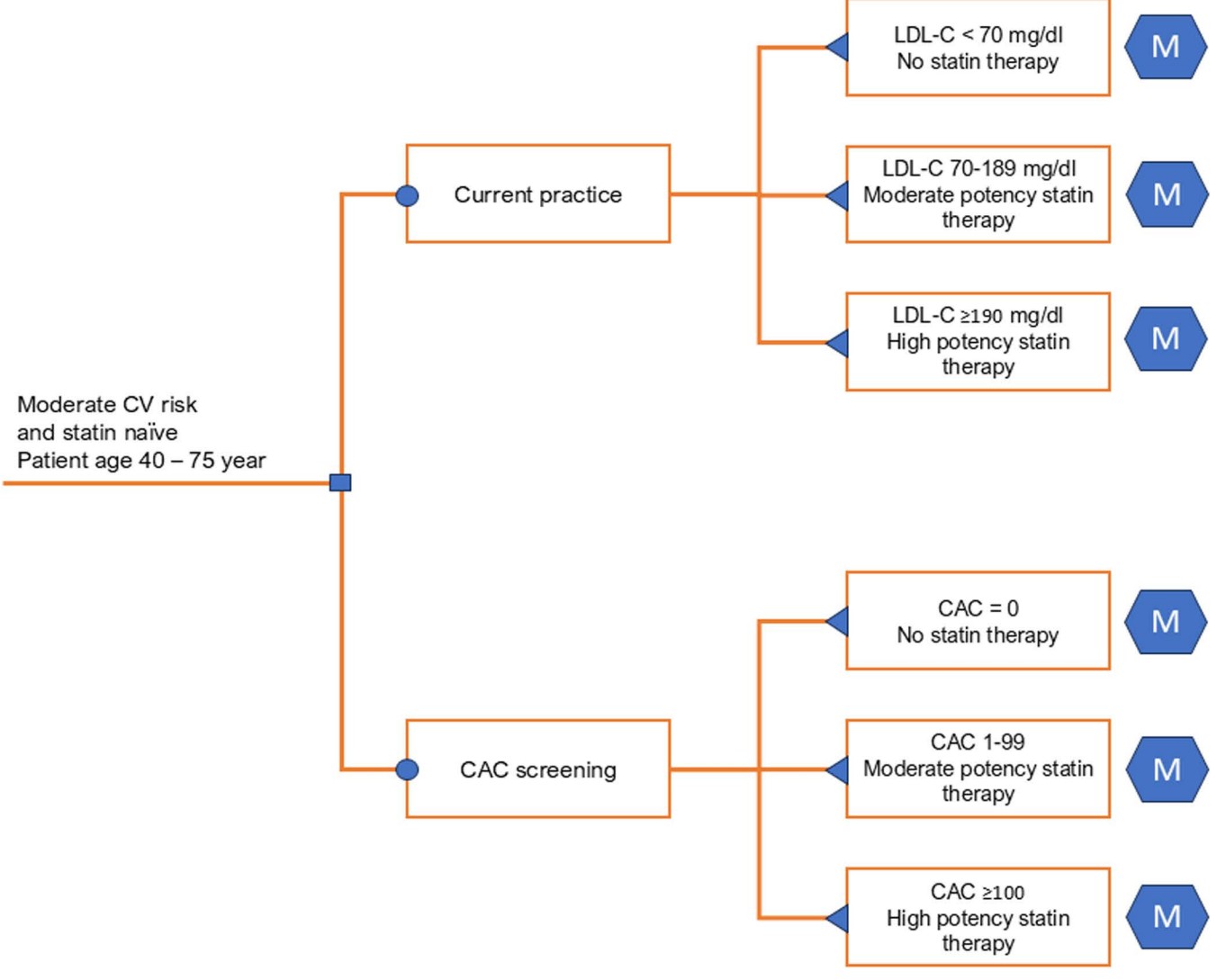

**Fig 1. Decision tree diagram showing two screening strategies.** Abbreviations: CAC, coronary artery calcium score; CV, cardiovascular; LDL-C, low-density lipoprotein cholesterol; "M" signs at the end of the decision tree, Markov model.

The probabilities of experiencing an acute non-fatal MI, acute non-fatal stroke, CVD death, and death from any cause were used to determine transitions to other health states in each one-year cycle. Patients in the post non-fatal MI state could experience a recurrent non-fatal MI, acute non-fatal stroke, or death, leading to transitions to the post recurrent non-fatal MI, post non-fatal MI and stroke, or dead state. Similar events could occur for patients in the post non-fatal stroke and post non-fatal MI and stroke states. After experiencing any CVD event, patients would be prescribed a high-potency statin. The model structures were reviewed and validated by cardiologists and health economists.

## Data sources

Prevalence, effectiveness of statin therapy, cost, and transition probabilities were retrieved from published literature and primary data collection, summarised in Tables 1 and 2. Since the aim of this analysis was to inform policymakers in

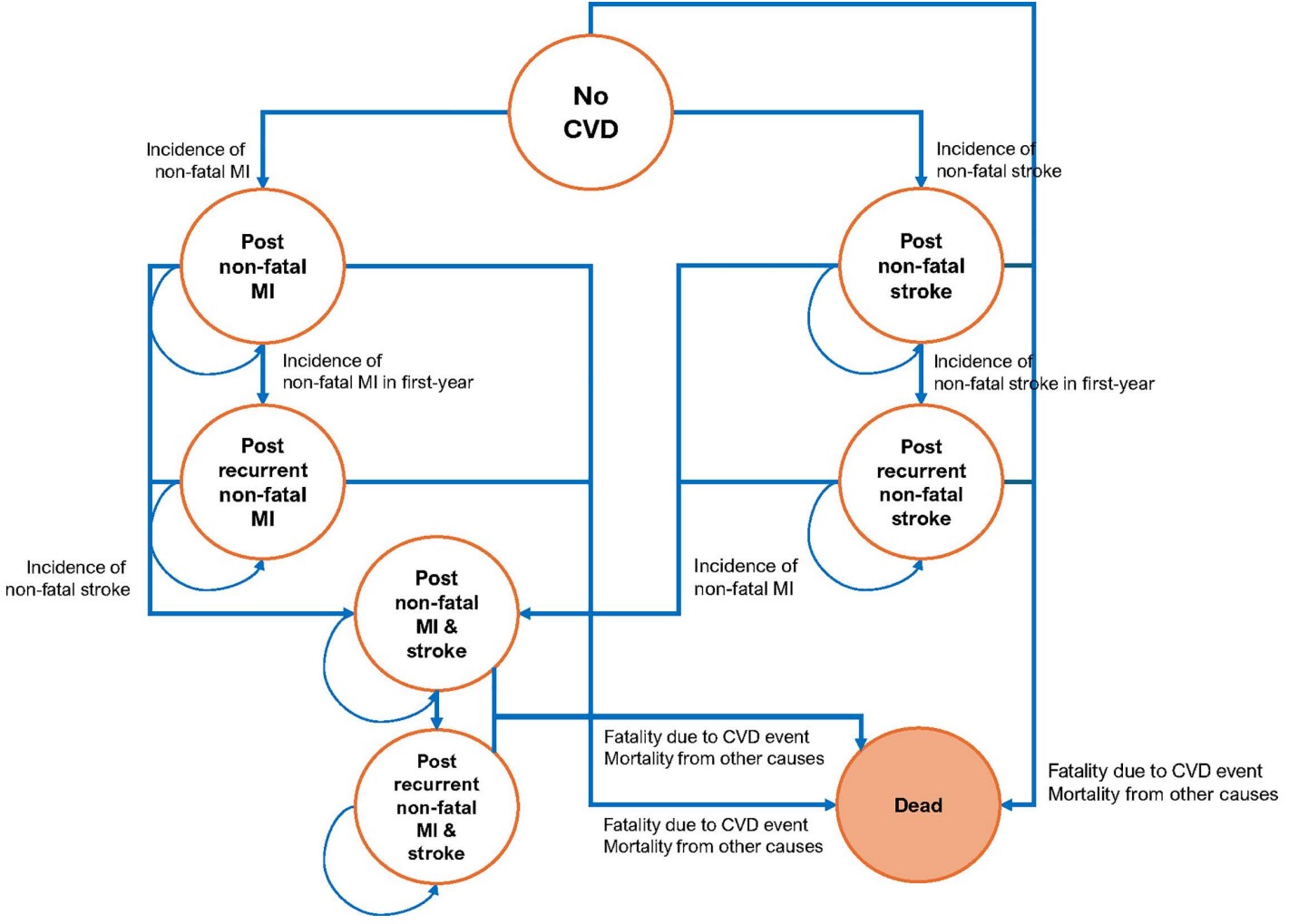

**Fig 2. State-transition diagram for Markov model.** Abbreviations: CVD, cardiovascular diseases; MI, myocardial infarction.

**Table 1. Selected input parameters for the decision tree model and corresponding treatment strategy.**

| Input parameter: probabilities | Base-case value mean (SE) | Receiving treatment |
|---|---|---|
| **Current practice strategy** | | |
| Incidence of initial LDL-C ≥ 190 mg/dL | 0.018 (0.013) | High potency statin |
| Incidence of initial LDL-C 70–189 mg/dL | 0.759 (0.04) | Moderate potency statin |
| Incidence of initial LDL-C < 70 mg/dL | 0.223 (0.04) | No statin |
| **CAC screening strategy** | | |
| Incidence of initial CAC ≥ 100 | 0.36 (0.013) | High potency statin |
| Incidence of initial CAC 1–99 | 0.39 (0.013) | Moderate potency statin |
| Incidence of initial CAC = 0 | 0.26 (0.012) | No statin |

**\*The input data were sourced from the hospital database from 2012 to 2022. A beta distribution was selected for these input parameters in the probabilistic sensitivity analysis.** Abbreviations: CAC, coronary artery calcium score; LDL-C, low-density lipoprotein cholesterol; SE, standard error.

**Table 2.  Selected input parameters for the Markov model.**

| Input parameters | Base-case value mean (SE) | Distribution | References |
|---|---|---|---|
| **Transitional probability of the CVD event following each reclassification groups** | | | |
| **No CVD state** | | | |
| Incidence of non-fatal MI in patient without statin therapy (Person-year) | 0.00558 (0.00039) | Beta | [13,29] |
| Incidence of non-fatal stroke in patient without statin therapy (Person-year) | 0.01348 (0.00106) | Beta | [13,29] |
| Incidence of CVD death in patient without statin therapy (Person-year) | 0.00148 (0.00013) | Beta | [13,29] |
| Incidence of non-fatal MI in patient with CAC = 0 (Person-year) | 0.00325 (0.00050) | Beta | [13] |
| Incidence of non-fatal MI in patient with CAC 1–99 (Person-year) | 0.00340 (0.00042) | Beta | [13] |
| Incidence of non-fatal MI in patient with CAC ≥ 100 (Person-year) | 0.00626 (0.00064) | Beta | [13] |
| Incidence of non-fatal stroke in patient with CAC = 0 (Person-year) | 0.00883 (0.00137) | Beta | [13] |
| Incidence of non-fatal stroke in patient with CAC 1–99 (Person-year) | 0.00918 (0.00114) | Beta | [13] |
| Incidence of non-fatal stroke in patient with CAC ≥ 100 (Person-year) | 0.01689 (0.00175) | Beta | [13] |
| Incidence of CVD death in patient with CAC = 0 (Person-year) | 0.00105 (0.00016) | Beta | [13] |
| Incidence of CVD death in patient with CAC 1–99 (Person-year) | 0.00103 (0.00014) | Beta | [13] |
| Incidence of CVD death in patient with CAC ≥ 100 (Person-year) | 0.00190 (0.00021) | Beta | [13] |
| **Non-fatal MI state** | | | |
| Probability of developing first-year recurrent non-fatal MI | 0.1000 (0.0010) | Beta | [30] |
| Probability of developing first-year non-fatal stroke | 0.0240 (0.0004) | Beta | [30] |
| Probability of developing first-year CVD death | 0.0810 (0.0010) | Beta | [30] |
| **Non-fatal stroke state** | | | |
| Probability of developing first-year non-fatal MI | 0.0095 (0.0004) | Beta | [31] |
| Probability of developing first-year recurrent non-fatal stroke | 0.0900 (0.0010) | Beta | [31] |
| Probability of developing first-year CVD death | 0.0340 (0.0040) | Beta | [31] |
| **CAC progression per 5 years in patient previously CAC = 0** | | | |
| 45–49 developing score 1–99, ≥ 100 | 0.341, 0.158 | | Hospital database 2012–2022 |
| 50–54 developing score 1–99, ≥ 100 | 0.338, 0.195 | | Hospital database 2012–2022 |
| 55–59 developing score 1–99, ≥ 100 | 0.331, 0.237 | | Hospital database 2012–2022 |
| 60–64 developing score 1–99, ≥ 100 | 0.319, 0.284 | | Hospital database 2012–2022 |
| 65–69 developing score 1–99, ≥ 100 | 0.302, 0.336 | | Hospital database 2012–2022 |
| 70–74 developing score 1–99, ≥ 100 | 0.281, 0.390 | | Hospital database 2012–2022 |
| **Statin efficacy** | | | |
| RR of moderate potency vs. no statin on incident of non-fatal MI | 0.76 (0.026) | Beta | [29] |
| RR of moderate potency vs. no statin on incident of non-fatal stroke | 0.86 (0.041) | Beta | [29] |

*(Continued)*

| Input parameters | Base-case value mean (SE) | Distribution | References |
|---|---|---|---|
| RR of moderate potency vs. no statin on incident of CVD death | 0.88 (0.020) | Beta | [29] |
| RR of high potency vs. no statin on incident of non-fatal MI | 0.58 (0.026) | Beta | [29] |
| RR of high potency vs. no statin on incident of non-fatal stroke | 0.74 (0.041) | Beta | [29] |
| RR of high potency vs. no statin on incident of CVD death | 0.77 (0.020) | Beta | [29] |
| **Average Cost per year (THB)** (approximately 36 THB = 1 US$ in 2024) | | | |
| **Direct Medical Cost** | | | |
| Moderate potency statin | 2,589 (259*) | Gamma | Hospital database 2021–2024 [32] |
| High potency statin | 9,071 (907*) | Gamma | Hospital database 2021–2024 [32] |
| CAC test | 4,000 (400*) | Gamma | [33] |
| Treatment of first-year non-fatal MI | 149,285 (68,107) | Gamma | [34] |
| Treatment of first-year recurrent non-fatal MI | 149,285 (68,107) | Gamma | [34] |
| Treatment of annual follow-up non-fatal MI | 29,340 (2,934) | Gamma | [34] |
| Treatment of first year non-fatal stroke | 191,467 (54,378) | Gamma | [35] |
| Treatment of first-year recurrent non-fatal stroke | 92,396 (7,670) | Gamma | [35] |
| Treatment of annual follow-up non-fatal stroke | 37,7812 (3,198) | Gamma | [35] |
| **Direct Non-medical Cost** | | | |
| Non-medical cost of first-year non-fatal MI patient | 3,667 (367*) | Gamma | [34] |
| Non-medical cost of annual follow-up non-fatal MI patient | 5,304 (530*) | Gamma | [34] |
| Non-medical cost of first-year non-fatal stroke patient | 72,309 (2279) | Gamma | [35] |
| Non-medical cost of annual follow-up non-fatal stroke patient | 46,390 (1,470) | Gamma | [35] |
| **Utility for each health state** | | | |
| Post non-fatal MI | 0.828 (0.038) | Beta | [36] |
| Post non-fatal stroke | 0.69 (0.062) | Beta | [36] |
| Post non-fatal stroke and MI | 0.678 (0.036) | Beta | [37] |
| Disutility due to recurrent non-fatal MI | 0.147 (0.015*) | Beta | [38] |
| Disutility due to recurrent non-fatal stroke | 0.226 (0.022*) | Beta | [38] |
| Disutility due to recurrent non-fatal MI or stroke in patient with previous non-fatal CVD event | 0.187 (0.019*) | Beta | [38] |

*Assume SE was 10% from mean value. The utility of No CVD health state was provided in S1 File, Table S4. **Abbreviations:** CAC, coronary artery calcium score; CT, computed tomography; CVD, cardiovascular disease; MI, myocardial infarction; RR, relative risk; SE, standard error.

Thailand, the identified sources were most relevant to the Thai context. When local data were not available, the most relevant international publications were used [29,30,36–38].

Due to the lack of nationally representative published evidence, the initial incidence of LDL-C levels and CAC scores was retrieved from a Chiang Mai University hospital database covering the period from 2012 to 2022 (Table 1). All data were retrieved from standard electronic medical records. Data were collected from June 2023 to August 2023. Patient identity was viewed only during data collection and was not recorded. Details on the baseline characteristics of the cohort are available in Table S2 in S1 File.

The assumptions applied in the Markov model are based on literature [13,29–38]. Since LDL-C levels change dynamically over time, the transition probability for the first CVD event in the current guideline was assumed to be the same regardless of LDL-C level. The benefits of statin therapy, in terms of risk reduction for MI, stroke, and CVD mortality, were obtained from meta-analyses and considered equal for men and women [29].

We employed a societal perspective, incorporating direct medical and non-medical costs (including transportation, food, accommodation, and opportunity costs) into the model. Direct medical costs comprised the average annual cost of statin therapy, non-contrast cardiac CT (once if CAC is present, and every five years if CAC is absent), first-year specific CVD event-related treatment, and annual follow-up for CVD event-related treatment. Costs for non-contrast cardiac CT (CAC screening) and statins were based on the national healthcare reimbursement rate in 2023 [33] and retail prices from online national drug information (NDI) in 2024 [32]. An average price per unit of each statin potency was calculated from prescribing proportions in the hospital database from 2021 to 2024 (Table S5 in S1 File). Cost data were converted to 2024 values using the Thai consumer price index (CPI) [39] and presented in Thai Baht (THB) (approximately 36 THB = 1 US$ in 2024) [40]. Future costs and QALYs beyond the first year were discounted by 3% annually.

## Cost-effectiveness analysis

The cost-effectiveness was evaluated in terms of incremental cost and incremental QALYs over a 35-year time horizon using base-case values and represented as the incremental cost-effectiveness ratio (ICER). The ICER was then compared against the official willingness-to-pay threshold in Thailand, which is set at 160,000 THB per QALY [27].

## Sensitivity analyses

A probabilistic sensitivity analysis (PSA) was conducted to assess the uncertainty of our model. Each ICER value was calculated based on a random value drawn from each parameter distribution, and this process was repeated for 1,000 simulations. A beta distribution was selected for probability and utility parameters, a log-normal distribution was used for risk ratio parameters, and a gamma distribution was used for all cost parameters. In cases where parameter standard error (SE) was unavailable, we assumed an SE of 10% of the mean values. The simulated ICERs were presented in cost-effectiveness acceptability curves (CEAC) for willingness-to-pay thresholds ranging from 0 to 600,000 THB/QALY to evaluate the probability that a strategy was cost-effective.

Beyond the PSA, we performed several additional sensitivity analyses to assess the impact of important assumptions. The following input parameters were evaluated for their ICER response by varying upper and lower range values and were presented using a tornado diagram: the incidence of initial LDL-C levels and CAC scores, transition probability of the first CVD event, statin price, treatment effectiveness, health state utilities, and discount rate (between 0% and 5%) [41].

For other types of sensitivity analyses, we performed scenario analyses by changing specific parameters of the model. First, we considered the effect of varying adherence to statin therapy in the current practice and CAC screening, based on awareness of their CAC score [41,42]. Adherence rates were considered as 19–52% [42–46] for the current practice strategy and 52% and 56% for CAC scores of 1–99 and ≥100, respectively [47]. Second, we evaluated the effect of changing the mean cost of high-potency statins to 15, 20, 30, and 35 THB per unit, and the mean cost of a CAC test to 2,000, 6,000, and 8,000 THB. Both interventions were also assessed for the value that exceeds the Thailand willingness-to-pay threshold. Third, we explored the impact of CAC progression by changing the CAC progression probability using a prediction model from the CAC Consortium, a large multicentre cohort of low-CV patients in the USA [48] (details on the probabilities by age are provided in Table S3 in S1 File).

Fourth, since we adopted the transition probabilities from Tiansuwan N. et al. [13], which included non-naïve statin patients and a significant proportion of DM patients, the baseline CV risk tends to be higher than in the general population. This may explain why patients with a CAC score of zero carried a similar risk to those with a CAC score of 1–99 (Table 2) and had a risk twice as high as reported in previous studies [19,21,24]. We considered reducing all CVD event rates in patients with a CAC score of zero to half of the base-case value. Lastly, we considered the effect of changing the incidence of initial CAC scores from Tiansuwan N.-derived incidences [13] to those of a lower-risk population. Therefore, we compared our base-case set with models that assumed incidence rates from the Multi-Ethnic Study of Atherosclerosis (MESA) cohort [11], a large, ongoing medical research study in the United States that aims to investigate the prevalence, causes, and progression of subclinical CVD in a diverse population. MESA, initiated in 2000, includes over 6,800 participants aged 45–84 from four major ethnic groups (White, African American, Hispanic, and Chinese American) who were free of cardiovascular disease at enrollment. The study's focus on advanced imaging, long-term follow-up, and diverse populations has significantly enhanced the understanding of early cardiovascular disease and improved risk prediction. The post-hoc sensitivity analysis using the healthcare provider's perspective was performed to account for the impact of direct non-medical costs.

## Results

### Base-case analysis

The comparison of total costs, life years, QALYs, and ICER from the base-case analyses is summarised in Table 3. The CAC screening strategy required an incremental cost of 10,091 THB to gain 0.62 QALYs per person over a 35-year time horizon, resulting in an ICER of 16,308 THB per QALY gained. Overall, the CAC screening strategy incurred higher costs but proved more effective.

### Sensitivity analyses

The PSA result showed that CAC screening became cost-effective at a willingness-to-pay threshold of 40,000 THB per QALY gained. At the Thai threshold of 160,000 THB per QALY gained, the probability of cost-effectiveness was 71% for CAC screening (Fig 3).

The tornado diagram highlighted the most influential parameter as the incidence of non-fatal stroke in patients without statin therapy. Variations in this parameter led to changes in the ICER from 146,112 THB to −32,522 THB, compared to the base-case ICER of 16,308 THB. The next most influential parameters included the cost of high-potency statins, the incidence of non-fatal stroke in patients with CAC scores of 1–99 and CAC ≥ 100, and the incidence of initial CAC scores ≥100 (Fig 4).

In scenario analyses, varying statin adherence assumptions (ranging from 19% to 52% in the current practice strategy versus approximately 55% in CAC screening) showed that CAC screening remained cost-effective at the Thai threshold of 160,000 THB per QALY gained (Table 4). Lower adherence to the current practice strategy favoured the

**Table 3. Costs, Utility, and Cost-effectiveness of each strategy in base-case analysis.**

|  | Current practice | CAC screening |
|---|---|---|
| Costs (THB) | 309,357 | 319,448 |
| Life years (year) | 19.37 | 19.88 |
| QALYs (year) | 16.55 | 17.17 |
| Incremental Costs (THB) | Reference | 10,091 |
| Incremental QALYs (year) | Reference | 0.62 |
| ICER per QALY gained | Reference | 16,308 |

**Approximately 36 THB = 1 US$ in 2024. Abbreviations:** CAC, coronary artery calcium score; ICER, incremental cost-effectiveness ratio; QALYs, Quality adjusted life years.

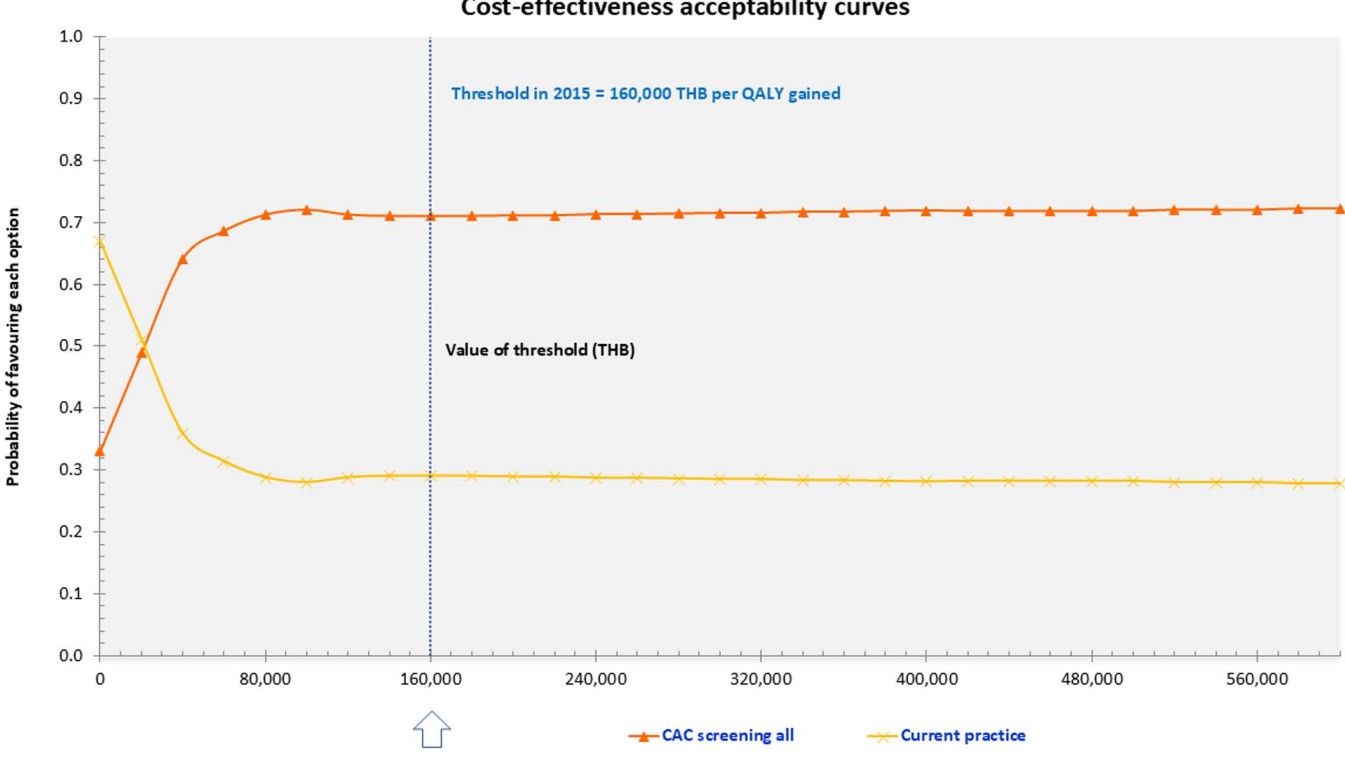

**Fig 3. Cost-effectiveness acceptability curve.** Abbreviations: CAC, coronary artery calcium score; QALYs, Quality adjusted life years.

cost-effectiveness of CAC screening. In terms of intervention cost, when the price of high-potency statins dropped to 20 THB per unit or less (Table 4), CAC screening was dominant. Conversely, rising high-potency statin prices led to increased incremental costs. When the annual cost of statin therapy was above 23,596 THB (equivalent to 65 THB per unit), CAC screening was not cost-effective.

A similar trend was observed with varying costs of the CAC test (Table 4), with the cost threshold being 92,028 THB per test. The results showed that slower CAC progression makes the CAC screening strategy more cost-effective, as fewer patients developed CAC scores of 1–99 and ≥100 (Table 4).

Decreasing the incidence of CVD events in patients with CAC = 0 by half of the base-case value made CAC screening more cost-effective (Table 4). Furthermore, simulations using initial CAC incidence data from the MESA cohort, where 50%, 26%, and 24% of the cohort had CAC scores of 0, 1–99, and ≥100, respectively, yielded similar conclusions (Table 4). The post-hoc sensitivity analysis using the healthcare provider's perspective demonstrated that CAC screening remains a cost-effective strategy (Table 4).

## Discussion

This study assessed the comparative cost-effectiveness of the CAC screening strategy and the current practice strategy. Our model indicated that the CAC screening strategy is cost-effective among patients with intermediate CV risk. The differences in cost and QALYs were minimal, with an ICER of 16,308 THB per QALY gained in the base-case analysis. At a willingness-to-pay threshold of 160,000 THB/QALY, CAC screening is likely to be cost-effective in 71% of simulations.

Based on our analysis, approximately one-third of patients who followed the CAC screening strategy received moderate- or high-potency statins at the beginning, while the majority of patients in the current practice received

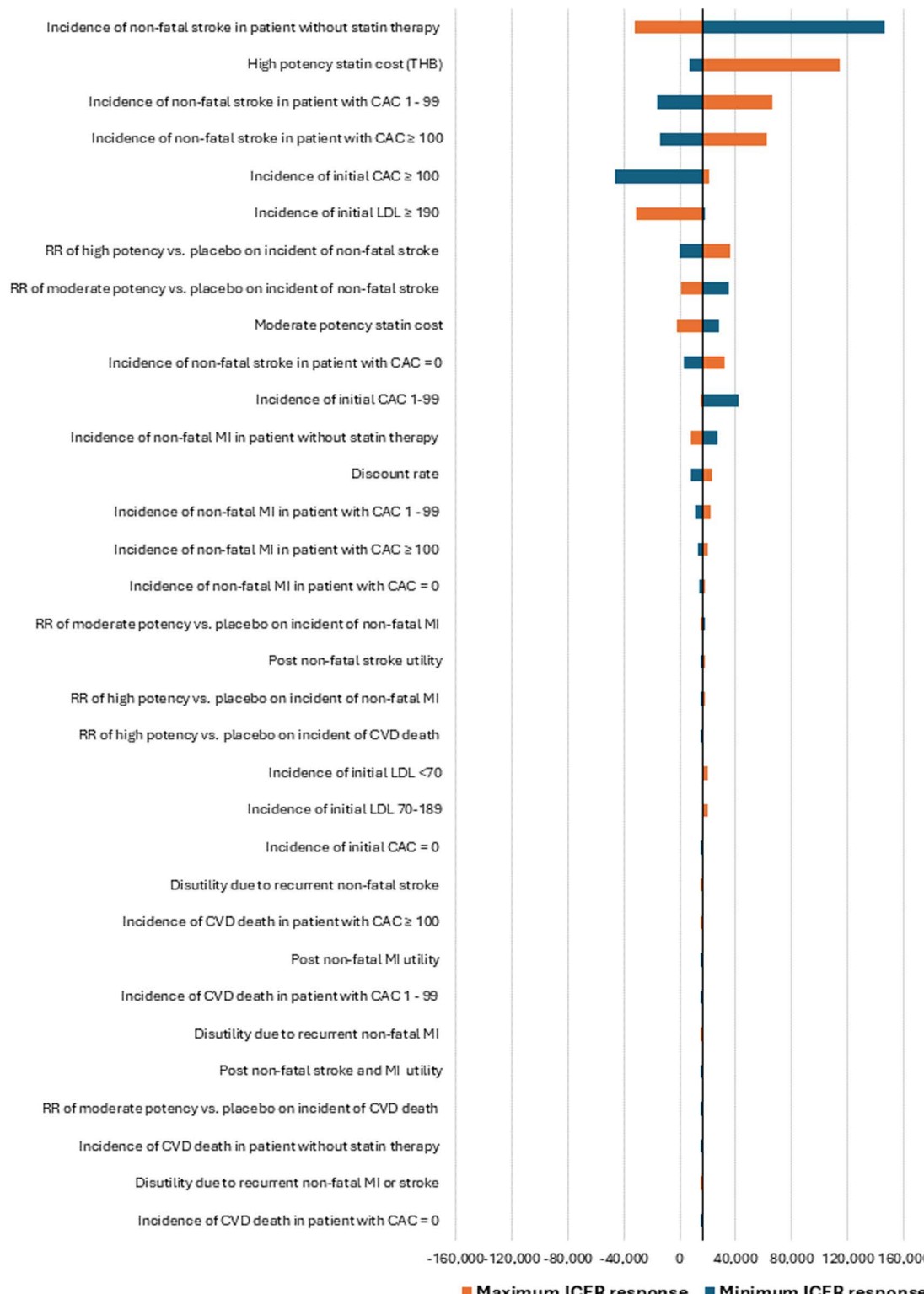

**Fig 4. Tornado diagram of one-way sensitivity analysis.** Discount rate was varied between 0 −5%. Abbreviations: CAC, coronary artery calcium score; CT, computed tomography; CVD, cardiovascular disease; LDL, low-density lipoprotein; MI, myocardial infarction; RR, relative risk; ICER, incremental cost-effectiveness ratio.

**Table 4. Sensitivity analysis based on multiple scenarios.**

| Scenario | | Incremental Cost (THB) | Incremental QALYs (years) | ICER | Decision* |
|---|---|---|---|---|---|
| **Drug adherence** | | | | | |
| Current practice | CAC screening | | | | |
| 19% | 36% for CAC =0 | -69,370 | 1.07 | -65,090 | CAC is dominant |
| 30% | 52% for CAC 1-99 | -45,636 | 0.88 | -51,834 | CAC is dominant |
| 52% | 56% for CAC ≥ 100 | 5,106 | 0.49 | 10,433 | CAC is cost-effective |
| 100% (base case) | 100% (base case) | 10,091 | 0.62 | 16,308 | CAC is cost-effective |
| **Cost parameter (THB)** | | | | | |
| High potency statin | CAC test | | | | |
| 5,479 (15 per unit) | 4,000 (base case) | -11,901 | 0.62 | -19,235 | CAC is dominant |
| 7,305 (20 per unit) | 4,000 (base case) | -722 | 0.62 | -1,167 | CAC is dominant |
| 9,071 (25 base case) | 4,000 (base case) | 10,091 | 0.62 | 16,308 | CAC is cost-effective |
| 10,956 (30 per unit) | 4,000 (base case) | 21,636 | 0.62 | 34,968 | CAC is cost-effective |
| 12784 (35 per unit) | 4,000 (base case) | 32,816 | 0.62 | 53,036 | CAC is cost-effective |
| 23,596 (64 per unit) | 4,000 (base case) | 98,999 | 0.62 | 160,000 | CAC is not cost-effective |
| | | | 0.62 | | |
| 9,071 (base-case) | 2,000 | 8,071 | 0.62 | 13,044 | CAC is dominant |
| 9,071 (base-case) | 4,000 (base case) | 10,091 | 0.62 | 16,308 | CAC is cost-effective |
| 9,071 (base-case) | 6,000 | 12,111 | 0.62 | 19,573 | CAC is cost-effective |
| 9,071 (base-case) | 8,000 | 14,131 | 0.62 | 22,838 | CAC is cost-effective |
| 9,071 (base-case) | 92,028 | 98,999 | 0.62 | 160,000 | CAC is not cost-effective |
| **Incident of CVD event in patient with CAC score of 0** | | | | | |
| Incidence of non-fatal MI (Person-year) | | | | | |
| 0.00160 | | 8,883 | 0.65 | 13,699 | CAC is cost-effective |
| 0.00325 (base-case) | | 10,091 | 0.62 | 16,308 | CAC is cost-effective |
| Incidence of non-fatal stroke (Person-year) | | | | | |
| 0.00440 | | -443 | 0.70 | -634 | CAC is dominant |
| 0.00883 (base-case) | | 10,091 | 0.62 | 16,308 | CAC is cost-effective |
| Incidence of non-fatal CVD death (Person-year) | | | | | |
| 0.00052 | | 10,340 | 0.64 | 16,284 | CAC is cost-effective |
| 0.00104 (base-case) | | 10,091 | 0.62 | 16,308 | CAC is cost-effective |
| **Incident of initial CAC score (Probability)** | | | | | |
| MESA cohort [11] | | | | | |
| CAC ≥ 100 | 0.24 | -4,902 | 0.48 | -10,136 | CAC is dominant |
| CAC 1-99 | 0.26 | | | | |
| CAC = 0 | 0.50 | | | | |
| Tiansuwan, N. (base-case) [7] | | | | | |
| CAC ≥ 100 | 0.36 | 10,091 | 0.62 | 16,308 | CAC is cost-effective |
| CAC 1-99 | 0.39 | | | | |
| CAC = 0 | 0.26 | | | | |
| **CAC progression parameter** | | | | | |
| Based on CAC Consortium [45] | | 13 | 0.64 | 20 | CAC is cost-effective |
| Based on Hospital data (base-case) | | 10,091 | 0.62 | 16,308 | CAC is cost-effective |
| **Perspective** | | | | | |
| Health care provider perspective | | 24,762 | 0.62 | 40,020 | CAC is cost-effective |
| Societal perspective (base-case) | | 10,091 | 0.62 | 16,308 | CAC is cost-effective |

*(Continued)*

**Table 4.** (Continued)

| Scenario | | Incremental Cost (THB) | Incremental QALYs (years) | ICER | Decision* |
|---|---|---|---|---|---|
| **Discounting rate** | | | | | |
| 0% | | 8,772 | 1.15 | 7,608 | CAC is cost-effective |
| 3% (base-case) | | 10,091 | 0.62 | 16,308 | CAC is cost-effective |
| 5% | | 10,165 | 0.44 | 23,352 | CAC is cost-effective |

*Cost-effectiveness was interpreted under the Thai willingness-to-pay threshold of 160,000 THB per QALY gained. Approximately 36 THB = 1 US$ in 2024. **Abbreviations:** CAC, coronary artery calcium score; ICER, incremental cost-effectiveness ratio; QALYs, Quality adjusted life years.

moderate-potency statins (76%). Moderate-potency statins appear to be the predominant potency in clinical practice according to the ACC/AHA guideline 2019 [9,43]. We found that when the annual cost of high-potency statins increased, CAC screening became more costly. Conversely, with the rising cost of moderate-potency statins, the CAC screening strategy became more cost-effective. This result demonstrated the differing impact of drug cost and its distribution among strategies. Since the cost of moderate-potency statins affects the current practice significantly more than CAC screening, an increase in moderate-potency statin costs leads to a relatively lower overall cost for CAC screening, making it more cost-effective. However, in most real-world situations, the cost of statins tends to be lower due to the country's health insurance coverage and drug reimbursement policies [49]. We found that when the cost of high-potency statins was below 20 THB, the CAC screening strategy became less costly and more effective.

Drug adherence also plays a key role in the cost-effectiveness of the strategy. Some patients prefer to avoid taking daily preventive medication [50]. Thus, treating all patients with statins may not be an appropriate strategy. In our analysis, higher adherence rates in the CAC screening group showed lower overall costs. With minimal costs for performing CAC testing, CAC scores can enhance the shared decision-making process through more accurate risk prediction. This helps reduce low-value pharmacological therapy and guides treatment decisions toward a patient-centred strategy [24]. Since the CAC test is not covered by universal coverage in our country, the price tends to be higher than the national healthcare reimbursement rate. However, we found that the CAC test would need to cost approximately 92,000 THB to render the CAC screening strategy not cost-effective at a willingness-to-pay threshold of 160,000 THB per QALY.

It is important to note that our simulated patients with a CAC score of zero had twice the risk of a CVD event compared to previous health economic studies [19,21,24]. The results of our model were also highly sensitive to the incidence of non-fatal stroke. Our analysis, which assumed a 50% reduction in CVD events from the base-case value in patients with a CAC score of zero, demonstrated a favourable outcome for CAC screening. In our base-case analysis, the proportion of the population placed on statins was similar for both the current practice and the CAC screening strategy (approximately 78% vs 75%). We also modelled the incidence of initial CAC scores based on the MESA cohort [11], which had a higher proportion of patients with a CAC score of zero, resulting in fewer patients being placed on statins. The simulation results similarly concluded that CAC screening is less costly and more effective. CAC screening allows for more efficient allocation of pharmacotherapy, requiring statin use in fewer patients to achieve the same reduction in events [17].

## Strengths and limitations

It was previously revealed that many patients who had non-fatal MI or stroke went on to experience other major hard events within a year [30,31]. In the real world, the transition probabilities, utility, and costs within post non-fatal MI and stroke states probably differ from other health states. Additionally, CAC scores naturally progress with age. Our study's strength lies in incorporating these health states and the CAC progression parameter into the Markov model, ensuring a more realistic representation of patient outcomes. Although it is difficult to compare with previous studies due

to different models, assumptions, and target populations, the demonstration of the dominant CAC screening strategy adds weight to prior studies [19–21,23,24]. Our study is the first to address a common clinical scenario for Thai patients and clinicians in deciding on the initiation of long-term statin therapy for patients at intermediate risk for CVD using the ACC/AHA cholesterol guideline 2019 [9]. Our findings indicated that the CAC screening strategy is probably more cost-effective and strengthens healthcare providers' ability to follow CVD prevention guidelines.

There are some limitations regarding the design and data used in this study. First, our analysis may underestimate costs as we did not include expenses related to patient time lost due to diagnostic testing or physician visits prior to a CVD event. Second, we did not account for potential cancer-related risks associated with CAC screening radiation. However, such complications are rare [51] and unlikely to significantly influence outcomes [19]. Third, we did not account for potential disutility and complications associated with statin use due to limited local data. We attempted to address this through scenario analysis, assuming varying levels of drug adherence among patients. In a scenario where there was a higher adherence rate in the CAC screening group, the strategy became dominant and less costly than the base case.

Fourth, some input parameters were derived from data specific to our centre, which may not be fully representative of the Thai population. We recommend rigorously collecting large national datasets for these parameters in future budget impact analyses before implementation. Finally, we did not consider the potential synergistic benefits of statins with other modalities such as anti-diabetic or anti-hypertensive therapies. The literature referenced in our study includes a mix of intermediate CV risk patients with comorbidities like diabetes mellitus and hypertension, which may overestimate the CVD event rate. Future research could incorporate the effects of these additional therapies into the model to reflect reality. However, our study focuses on statin-naïve patients who may be healthier than those simulated in our model. In our scenario analysis, which assumed a 50% reduction in CVD events from the base-case value for patients with a CAC score of zero, the CAC screening strategy demonstrated a reduction in incremental cost, making it a dominant strategy.

## Conclusion

The CAC screening strategy as part of ASCVD primary prevention among Thai patients with intermediate risk is probably cost-effective at a willingness-to-pay threshold of 160,000 THB per QALY gained. CAC screening appears to be the dominant strategy under a wide range of scenarios, especially when the costs of CAC screening and high-potency statins are low. Our study highlights the potential for implementing CAC screening in the Thai context. However, further budget impact analysis studies should be conducted to assess the affordability of healthcare technologies when making policy decisions.

## Supporting information

**S1 File. Model summary.**
(DOCX)

**S1 Fig. Cost-effectiveness plane of probabilistic sensitivity analysis.**
(PNG)

## Acknowledgments

This study was partially supported by the Faculty of Medicine, Chiang Mai University and Chiang Mai University itself, the University of Phayao, and the Thailand Science Research and Innovation Fund. We extend our deep gratitude to cardiologists Songsak Kiatchoosakun, Suphot Srimahachota, Arintaya Phrommintikul, and Yotsawee Chotechuang for their valuable review and validation of the model. Additionally, we thank Wilarat Saiyarat, a pharmacist at Chiang Mai University Hospital, for providing essential information on input parameters.

## Author contributions

**Conceptualization:** Pakpoom Wongyikul, Phichayut Phinyo, Pannipa Suwannasom, Apichat Tantraworasin, Surasak Saokaew.

**Data curation:** Pakpoom Wongyikul, Phichayut Phinyo, Surasak Saokaew.

**Formal analysis:** Pakpoom Wongyikul, Phichayut Phinyo, Surasak Saokaew.

**Investigation:** Pakpoom Wongyikul, Phichayut Phinyo, Pannipa Suwannasom, Apichat Tantraworasin, Surasak Saokaew.

**Methodology:** Pakpoom Wongyikul, Phichayut Phinyo, Pannipa Suwannasom, Apichat Tantraworasin, Surasak Saokaew.

**Project administration:** Phichayut Phinyo.

**Resources:** Pakpoom Wongyikul, Phichayut Phinyo, Pannipa Suwannasom, Apichat Tantraworasin, Surasak Saokaew.

**Software:** Pakpoom Wongyikul, Phichayut Phinyo, Surasak Saokaew.

**Supervision:** Phichayut Phinyo.

**Validation:** Pakpoom Wongyikul, Phichayut Phinyo, Pannipa Suwannasom, Apichat Tantraworasin, Surasak Saokaew.

**Writing – original draft:** Pakpoom Wongyikul.

**Writing – review & editing:** Pakpoom Wongyikul, Phichayut Phinyo, Pannipa Suwannasom, Apichat Tantraworasin, Surasak Saokaew.

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
