## [Decision Letter · Decision Letter 0]

17 Sep 2024

Dear Dr. Phinyo,

We look forward to receiving your revised manuscript.

Kind regards,

Vijay S. Gc, PhD

Academic Editor

PLOS ONE

Journal Requirements:

1. When submitting your revision, we need you to address these additional requirements. Please ensure that your manuscript meets PLOS ONE's style requirements, including those for file naming. The PLOS ONE style templates can be found at https://journals.plos.org/plosone/s/file?id=wjVg/PLOSOne_formatting_sample_main_body.pdf and https://journals.plos.org/plosone/s/file?id=ba62/PLOSOne_formatting_sample_title_authors_affiliations.pdf 2. We suggest you thoroughly copyedit your manuscript for language usage, spelling, and grammar. If you do not know anyone who can help you do this, you may wish to consider employing a professional scientific editing service.  The American Journal Experts (AJE) (https://www.aje.com/) is one such service that has extensive experience helping authors meet PLOS guidelines and can provide language editing, translation, manuscript formatting, and figure formatting to ensure your manuscript meets our submission guidelines. Please note that having the manuscript copyedited by AJE or any other editing services does not guarantee selection for peer review or acceptance for publication.  Upon resubmission, please provide the following: The name of the colleague or the details of the professional service that edited your manuscript A copy of your manuscript showing your changes by either highlighting them or using track changes (uploaded as a *supporting information* file) A clean copy of the edited manuscript (uploaded as the new *manuscript* file)” 3. We note that there is identifying data in the Supporting Information file <Supplementary material.docx>. Due to the inclusion of these potentially identifying data, we have removed this file from your file inventory. Prior to sharing human research participant data, authors should consult with an ethics committee to ensure data are shared in accordance with participant consent and all applicable local laws. Data sharing should never compromise participant privacy. It is therefore not appropriate to publicly share personally identifiable data on human research participants. The following are examples of data that should not be shared: -Name, initials, physical address-Ages more specific than whole numbers-Internet protocol (IP) address-Specific dates (birth dates, death dates, examination dates, etc.)-Contact information such as phone number or email address-Location data-ID numbers that seem specific (long numbers, include initials, titled “Hospital ID”) rather than random (small numbers in numerical order) Data that are not directly identifying may also be inappropriate to share, as in combination they can become identifying. For example, data collected from a small group of participants, vulnerable populations, or private groups should not be shared if they involve indirect identifiers (such as sex, ethnicity, location, etc.) that may risk the identification of study participants. Additional guidance on preparing raw data for publication can be found in our Data Policy (https://journals.plos.org/plosone/s/data-availability#loc-human-research-participant-data-and-other-sensitive-data) and in the following article: http://www.bmj.com/content/340/bmj.c181.long. Please remove or anonymize all personal information (<specific identifying information in file to be removed>), ensure that the data shared are in accordance with participant consent, and re-upload a fully anonymized data set. Please note that spreadsheet columns with personal information must be removed and not hidden as all hidden columns will appear in the published file. 4. Please include captions for your Supporting Information files at the end of your manuscript, and update any in-text citations to match accordingly. Please see our Supporting Information guidelines for more information: http://journals.plos.org/plosone/s/supporting-information.

Reviewers' comments:

Reviewer's Responses to Questions

**Comments to the Author**

1. Is the manuscript technically sound, and do the data support the conclusions?

Reviewer #1: No

Reviewer #2: Yes

2. Has the statistical analysis been performed appropriately and rigorously?

Reviewer #1: No

Reviewer #2: Yes

3. Have the authors made all data underlying the findings in their manuscript fully available?

Reviewer #1: Yes

Reviewer #2: Yes

4. Is the manuscript presented in an intelligible fashion and written in standard English?

Reviewer #1: Yes

Reviewer #2: Yes

Reviewer #1: The authors aim to evaluate the cost-effectiveness of using Coronary Artery Calcium (CAC) screening as an alternative to the current screening guidelines in Thailand. However, I recommend that this study be rejected for publication in PLOS ONE.

The primary concern is that the relative effect between CAC screening and the current guidelines has not been appropriately estimated. There are two key issues:

Initial Incidence Data: The initial incidence of LDL-C and CAC levels is derived from only 112 patients at a single hospital. This small, non-representative sample cannot adequately reflect the Thai population. Additionally, the inclusion of patients who must have CAC testing introduces selection bias.

Long-Term Effect Estimation: The long-term effect of CAC screening is based on the estimated progression of CAC by age, derived from a retrospective study without a direct comparator. The progression data for comparator is sourced from another trial, making the relative effect unreliable.

This flaw is fundamental to the cost-effectiveness analysis and cannot be resolved through revision; substantial additional data on the relative effect is required.

Reviewer #2: Thank you for the thoughtful research. Here are some specific and general comments:

Abstract:

-The statement in the methods “Different potencies of statin were initiated based on CAC score and ACC/AHA guidelines 2019 recommendation” is not clear to me. From the introduction my understanding is that this was comparing CAC screening plus treatment to standard treatment as directed by guidelines. Without reading the body of the paper, it’s confusing how many different interventions are being compared.

-Stating that the old practice has a 29% of being cost-effective compared to the CAC intervention having a probability of 71% is redundant; in a comparison of two interventions the probability that the other is cost-effective is going to be 100-(probability other is cost-effective). This could be removed for space and replaced with other information.

-Recommend rounding all dollar figures, including ICERs, to the nearest whole number (i.e. no decimal places). What is the Thai WTP threshold? It might be useful to provide a conversion to USD in the abstract for international readers.

Introduction:

-Perhaps provide a short definition in lay terms of dyslipidemia and atherosclerosis.

-What is meant by “subclinical”

-Suggest modifying the sentence: “This study aims to evaluate the cost-effectiveness of CAC screening for primary prevention in Thai patients with intermediate ASCVD risk, compared to the current guidelines according to the ACC/AHA guidelines 2019 recommendation” to be clearer. Propose something like: “This study aims to evaluate the cost-effectiveness of including CAC screening for primary prevention in Thai patients with intermediate ASCVD risk as per the ACC/AHA guidelines 2019 recommendation, compared to the current Thai guidelines which do not advise screening in this population.”

Model Overview:

-Revise first sentence “A hybrid model combining a decision tree and a hybrid model combining a decision tree and Markov model was constructed (Fig 1 and 2)” – sounds like “hybrid model combining a decision tree” is repeated?

- “Quality-adjusted life-years (QALYs) were used as the outcome utility measure.” I think you can delete “utility” and just say outcome measure.

-Can a reference be provided for the Thai guidelines?

Data Sources

-“Patient identity was access only during the data collection and was not collected.” Suggest “viewed only during data collection and was not recorded”

Table 1: since all have the same distribution and source, these columns could be combined/moved to a footnote of the table.

Table 2: Personally I think five decimal places is a lot, I think displaying four in the table would be OK. As commented for the abstract rounding costs to whole number is advised. What is the utility for the “No CVD” health state?

Sensitivity Analysis:

-What about a scenario analysis where the discount rate is varied (or set to zero)?

-“MESA” acronym – needs to be defined and described

Results

Table 4: suggest revising the title, it is currently not clear/grammatically correct. I don’t think stating something is “more dominant” or “less dominant” is appropriate. CAC is not dominant in the base case. I think just stating whether it is or is not dominant is appropriate.

Discussion

-Using “dominant” to describe the results is not appropriate. To me this terminology is only used when an intervention is both less costly and more effective. The CAC strategy is cost-effective but not dominant. If it becomes dominant in a scenario analysis that can be stated, but stating “more dominant” or “less dominant”, does not align with my understanding of the base case findings.

-It's interesting to me that the cost of the statins seems to be driving the higher cost of the CAC strategy. The way the discussion phrases it, it doesn’t seem like there is a benefit to the higher dose of statins, that it is simply an extra cost to be rationalized but doesn’t lead to avoidance of downstream costs or increase in utility. Is this the intention of the discussion section on statin costs?

-Is there any research on cost-effectiveness of CAC screening from other settings where this has already been included in guidelines?

Other comments:

Figure 4 needs abbreviations added. Also, the maximum and minimum ICER bars all being to one side of the base case does not make sense to me. How can this be correct? I’ve never seen a tornado diagram do that.

**Do you want your identity to be public for this peer review?** For information about this choice, including consent withdrawal, please see our Privacy Policy

Reviewer #1: No

Reviewer #2: **Yes: ** Rebecca Hancock-Howard

---

## [Author Response · Author response to Decision Letter 1]

21 Oct 2024

Reviewers comment on the manuscript

Entitles: “Cost-utility analysis of Coronary Artery Calcium screening to guide statin prescription among intermediate risk patient in Thailand”

Dear Editor and reviewers,

We would like to thank you for your valuable reviews and comments. It is our great pleasure to have an opportunity to revise our manuscript. We have revised and modified our manuscript with some additional information (track change) as suggested by reviewers’ comment. We hope that our revisions will improve the quality of the manuscript and give a clearer vision of research methodology to meet qualification for publication in PLOS ONE. Please inform us if further information or clarification needs to be addressed. We are looking forward to your reviews and would be extremely grateful to make our response.

Reviewer #1: The authors aim to evaluate the cost-effectiveness of using Coronary Artery Calcium (CAC) screening as an alternative to the current screening guidelines in Thailand. However, I recommend that this study be rejected for publication in PLOS ONE.

The primary concern is that the relative effect between CAC screening and the current guidelines has not been appropriately estimated. There are two key issues:

1. Initial Incidence Data: The initial incidence of LDL-C and CAC levels is derived from only 112 patients at a single hospital. This small, non-representative sample cannot adequately reflect the Thai population. Additionally, the inclusion of patients who must have CAC testing introduces selection bias.

Answer: We appreciated your feed back and would like to address your concern. We acknowledge that our data may not fully represent the true population of our country. However, the included patients were rigorously screened to ensure they were at intermediate cardiovascular risk and had never received statin therapy. We specifically selected patients who underwent CAC screening because our goal was to assess LDL distribution in individuals for whom CAC might be used to guide statin therapy.

We have provided incidence data for LDL categories based on previous evidence that closely aligns with our target patient population.

Table 1.

Study Country ASCVD 10-year risk Incidence distribution Inclusion criteria Disadvantage

Kevin E Kip

2024 [1] USA Intermediate Total N =32,801

LDL < 70 = 8.8 %

LDL 70-189 = 88.6 %

LDL >= 190 = 2.6% patients aged 50–89 years from tertiary care

and was restricted to primary prevention

(not on statin therapy at baseline or within 1year of follow-up.) Other country

Matthew E Gold

2020 [2] USA Mixed Total N =4,623,851

LDL >= 190 = 2.9% included adults (age ≥18 years) with LDL-C ≥190 mg/dL, at least one LDL-C level drawn from 255 health systems participatin Other country

Mixed ASCVD risk

Mixed population of patient receive statin

Tainsuwan N2023 [3]

Thai intermediate Total N= 1,427

LDL Mean ±SD

127.28 ±(36.8)

Patient underwent CAC without previous MACE event Mixed population of patient receive statin

Aekplakorn W

2009 [4] Thai Mixed Total N= 19,021

LDL Mean (SE)

128.7 (1.1) NHES 2009 Mixed population of patient receive statin

This study Thai intermediate Total N= 112

LDL median (IQR)

106 (81, 135)

LDL < 70 = 22.8 %

LDL 70-189 = 75.9 %

LDL >= 190 = 1.8% Chiang Mai University Hospital database from 2012 -2022

Patients were free of CVD, and not started statin therapy. Small sample size

Our data might be the best available data that align with our target population at the current time. The true LDL-C levels of population could be higher or lower than the values used in the base case. Sensitivity analyses were then conducted to account for uncertainties in these parameters. In the one-way sensitivity analysis, the incidence of initial LDL levels at 70 and 70-189 mg/dL showed minimal impact on the ICER, while LDL levels above 190 mg/dL emerged as a sensitive input parameter. We varied the probability of this incidence between 0% and 5% (the upper or lower limit were based on standard error), and the ICER demonstrated a trend toward negative values as the incidence increased (see red box in Figure 1). Importantly, the ICER did not exceed the willingness-to-pay (WTP) threshold of 160,000 THB when 0 was input, indicating that the conclusion for this parameter remains robust. However, we encourage the use of national data that specifically focus on collecting these input parameters in future budget impact analyses before implementation.

We have added the text regarding to this limitation in line 341-344 as follow: “Fourth, some input parameters were derived from data specific to our center with relatively have small sample size, which may not be fully representative of the Thai population. However, sensitivity analysis accounted for this parameter uncertainties showed robust outcome. We recommend rigorously collecting large national datasets for these parameters in future budget impact analyses before implementation.”

Figure 1. Tornado diagram of one-way sensitivity analysis

Reference

1. Kip KE, Diamond D, Mulukutla S, et al. Is LDL cholesterol associated with long-term mortality among primary prevention adults? A retrospective cohort study from a large healthcare system. BMJ Open2024;14:e077949. doi:10.1136/bmjopen-2023-077949

2. Gold, M. E., Nanna, M. G., Doerfler, S. M., Schibler, T., Wojdyla, D., Peterson, E. D., & Navar, A. M. (2020). Prevalence, treatment, and control of severe hyperlipidemia. American journal of preventive cardiology, 3, 100079. https://doi.org/10.1016/j.ajpc.2020.100079

3. Tiansuwan, N., Sasiprapha, T., Jongjirasiri, S., Unwanatham, N., Thakkinstian, A., Laothamatas, J., & Limpijankit, T. (2023). Utility of coronary artery calcium in refining 10-year ASCVD risk prediction using a Thai CV risk score. Frontiers in cardiovascular medicine, 10, 1264640. https://doi.org/10.3389/fcvm.2023.1264640

4. Aekplakorn W, Taneepanichskul S, Kessomboon P, et al. Prevalence of Dyslipidemia and Management in the Thai Population, National Health Examination Survey IV, 2009. J Lipids. 2014;2014:249584. doi:10.1155/2014/249584

2. Long-Term Effect Estimation: The long-term effect of CAC screening is based on the estimated progression of CAC by age, derived from a retrospective study without a direct comparator. The progression data for comparator is sourced from another trial, making the relative effect unreliable.

Answer: We appreciated your feedback and would like to address your concern. Our model evaluated the impact of statin initiation for primary prevention based on the 2019 ACC/AHA guideline with CAC screening compared to without CAC screening over a 35-year time horizon. CAC progression is an input parameter used to predict the likelihood of progression to CAC 1-99 or >100 in patients who initially have no detectable CAC. This parameter was not addressed in previous models [1-5], which assumed the CAC score remained constant over time. To account for the natural progression of the disease, we included this parameter in the model to construct a more realistic cost-effectiveness analysis and to ensure fairness for patients who initially have no detectable CAC.

The observational studies that assess long-term CAC progression in Thailand is limited, as these types of studies require considerable time and resources. Estimating CAC progression every five years using retrospective data from the Chiang Mai University Hospital database (2012-2022, N=112) may be the most relevant evidence available for our target population.

The efficacy of moderate- or high-potency statins for reduction the risk of major cardiovascular events is based on an average pooled effect from meta-analysis of multiple trials, with a median follow-up of up to 5 years [6]. As a result, we assumed that the long-term effect of statin therapy remains constant across all age groups.

Additionally, we conducted scenario analysis using CAC progression data estimated from a prediction model based on the CAC Consortium [7], a multicenter cohort study involving four high-volume centers in the United States (Total N= 22,346 patients) (Table 2), and a scenario where CAC progression parameters were not used in the model. The results showed that slower CAC progression makes the CAC screening strategy more cost-effective, as fewer patients developed CAC 1-99, and ≥ 100 (Table 3). However, the probability estimates from the CAC Consortium may not be proper to be use in base-case, as they are based on a population not closely aligned with our target group.

We have added this analysis in our appendix and addressed this issue as our strength as follow:

Method section: in line 207-210

“Third, we explored the impact of CAC progression by changing the CAC progression probability using prediction model from the CAC Consortium, a large multicenter cohort of low CV patients in the USA [45] (details on the probabilities by age are provided in Supplementary Table S3).”

Result section: in line 263-265

“The results showed that slower CAC progression makes the CAC screening strategy more cost-effective, as fewer patients developed CAC 1-99, and ≥ 100 (Table 4).”

Discussion section: in line 320-325

“It was previously revealed that many patients who had non-fatal MI or stroke went on to experience other major hard events within a year [27, 28]. In the real world, the transition probabilities, utility, and costs within post non-fatal MI & stroke probably differ from other health states. Additionally, CAC scores naturally progress with age. Our study's strength lies in incorporating these health states and the CAC progression parameter into the Markov model, ensuring a more realistic representation of patient outcomes.”

Table 2. Comparison of the probability of CAC progression over age among patient who initially have no detectable CAC

Baseline characteristic Hospital data based 2012-2022 CAC Consortium [7]

Age (years) Mean± SD: 63.4±7.62

Intermediate risk: 100%

CAC score categories (%): 0 (25.7), 1-99 (31.2), ≥100 (43.1) Age (years) Mean± SD: 43.5±4.5

Intermediate risk: 2.6%

CAC score categories (%): 0 (65.6), 1-99 (27.2), ≥100 (7.2)

Age Probability of having CAC

1-99, ≥ 100 Probability of having CAC

≥ 0 Probability of having CAC

≥ 0

45 – 49 0.341, 0.158 0.50 0.17

50 – 54 0.338, 0.195 0.53 0.26

55 – 59 0.331, 0.237 0.57 0.38

60 – 64 0.319, 0.284 0.60 0.50

65 – 69 0.302, 0.336 0.64 0.63

70 – 74 0.281, 0.390 0.67 0.74

Scenario Incremental

Cost (THB) Incremental QALYs (years) ICER Decision*

CAC progression parameter

Absence -6,866 0.62 -11,014 CAC is dominant

Based on CAC Consortium [7] 13 0.64 20 CAC is cost-effective

Based on Hospital data 10,091 0.62 16,308 CAC is cost-effective

Table 3. Scenario analysis

*Cost-effectiveness was interpreted under the Thai willingness-to-pay threshold of 160,000 THB per QALY gained. Approximately 36 THB = 1 US$ in 2024. Abbreviation: CAC, coronary calcium score; ICER, incremental cost-effectiveness ratio; QALYs, Quality adjusted life years.

Reference

1. van Kempen BJ, Spronk S, Koller MT, et al. Comparative effectiveness and cost-effectiveness of computed tomography screening for coronary artery calcium in asymptomatic individuals. J Am Coll Cardiol 2011;58:1690–701.

2. Roberts ET, Horne A, Martin SS, et al. Cost-effectiveness of coronary artery calcium testing for coronary heart and cardiovascular disease risk prediction to guide statin allocation: the Multi-Ethnic Study of Atherosclerosis (MESA). PLoS One 2015;10:e0116377.

3. van Kempen BJ, Ferket BS, Steyerberg EW, Max W, Myriam Hunink MG, Fleischmann KE. Comparing the cost-effectiveness of four novel risk markers for screening asymptomatic individuals to prevent cardiovascular disease (CVD) in the US population. Int J Cardiol 2016;203: 422–31.

4. Hong JC, Blankstein R, Shaw LJ, Padula WV, Arrieta A, Fialkow JA, Blumenthal RS, Blaha MJ, Krumholz HM, Nasir K. Implications of Coronary Artery Calcium Testing for Treatment Decisions Among Statin Candidates According to the ACC/AHA Cholesterol Management Guidelines: A Cost-Effectiveness Analysis. JACC Cardiovasc Imaging. 2017 Aug;10(8):938-952. doi: 10.1016/j.jcmg.2017.04.014. PMID: 28797417.

5. Galper BZ, Wang YC, Einstein AJ. Strategies for primary prevention of coronary heart disease based on risk stratification by the ACC/ AHA lipid guidelines, ATP III guidelines, coronary calcium scoring, and C-reactive protein, and a global treat-all strategy: a comparativeeffectiveness modeling study. PLoS One 2015; 10:e0138092.

6. Cholesterol Treatment Trialists' (CTT) Collaborators, Mihaylova, B., Emberson, J., Blackwell, L., Keech, A., Simes, J., Barnes, E. H., Voysey, M., Gray, A., Collins, R., & Baigent, C. (2012). The effects of lowering LDL cholesterol with statin therapy in people at low risk of vascular disease: meta-analysis of individual data from 27 randomised trials. Lancet (London, England), 380(9841), 581–590. https://doi.org/10.1016/S0140-6736(12)60367-5

7. Dzaye, O., Razavi, A. C., Dardari, Z. A., Shaw, L. J., Berman, D. S., Budoff, M. J., Miedema, M. D., Nasir, K., Rozanski, A., Rumberger, J. A., Orringer, C. E., Smith, S. C., Jr, Blankstein, R., Whelton, S. P., Mortensen, M. B., & Blaha, M. J. (2021). Modeling the Recommended Age for Initiating Coronary Artery Calcium Testing Among At-Risk Young Adults. Journal of the American College of Cardiology, 78(16), 1573–1583. https://doi.org/10.1016/j.jacc.2021.08.019

Reviewer #2: Thank you for the thoughtful research. Here are some specific and general comments:

Abstract:

1. The statement in the methods “Different potencies of statin were initiated based on CAC score and ACC/AHA guidelines 2019 recommendation” is not clear to me. From the introduction my understanding is that this was comparing CAC screening plus treatment to standard treatment as directed by guidelines. Without reading the body of the paper, it’s confusing how many different interventions are being compared.

Answer: In our study, statin therapy was initiated at either high or moderate potency based on the severity of each individual’s cardiovascular risk. Recent studies show that the CAC score improves cardiovascular risk assessment [1], helping to better identify patients needing intensive statin therapy or those who may not benefit. Our aim is to compare the cost-effectiveness of incorporating the CAC score as a cardiovascular risk assessment tool, based on the 2019 ACC/AHA guidelines, against not using it for the primary prevention of ASCVD. For the CAC screening strategy, individual patients are assessed based on their CAC score to guide statin therapy. A moderate-potency statin is prescribed for patients with a CAC score of 1-99, while a high-potency statin is recommended for those with a CAC score of ≥100. Patients with a CAC score of zero are advised to reassess their score every five years until a positive CAC score is detected.

Under the 2019 ACC/AHA guidelines, without using the CAC score, patients with low-density lipoprotein cholesterol (LDL-C) levels between 70 and 189 mg/dL are prescribed a moderate-potency statin, while those with LDL-C levels of 190 mg/dL or higher are prescribed a high-potency statin.

For more clarity, we have expanded the detailed on the intervention in line 35-47 as follow: “Objective: This study aims to evaluate the cost-utility of Coronary Artery Calcium (CAC) screening for primary prevention in Thai patients with intermediate cardiovascular diseases (CVD) risk, compared to the current practice according to the ACC/AHA guideline 2019 recommendation without using of CAC score.

Methods: A hybrid model combining a decision tree and Markov model was constructed to compare cost and QALYs from a societal perspective. The model evaluated the target population of statin-naïve individuals aged 40-75 with intermediate CVD risk. We assessed the impact of statin initiation for primary prevention based on the ACC/AHA guideline 2019 with CAC screening compared to without CAC screening over a 35-year time horizon.”

References

1. Tiansuwan, N., Sasiprapha, T., Jongjirasiri, S., Unwanatham, N., Thakkinstian, A., Laothamatas, J., & Limpijankit, T. (20

---

## [Decision Letter · Decision Letter 1]

28 Feb 2025

We look forward to receiving your revised manuscript.

Kind regards,

Andreas Zirlik, MD

Academic Editor

PLOS ONE

**Additional Editor Comments:**

There are numerous instances of incorrect grammar, it would be an enormous effort to correct them all as a reviewer. I note that the journal does not copyedit submissions but this should be done before proceeding with publication.

Here are more substantive comments:

Abstract: Should read “$160,00 BAHT per QALY” not QALYs

The statement: “Probabilistic, and additional one-way sensitivity analyses were performed to account for the model’s important assumption and robustness” is not clear. Suggest revising to say it is assessing robustness and testing the impact of assumptions.

Suggest rounding all monetary figures to the nearest whole value (i.e. remove decimals) throughout text.

Introduction line 69 and Methods line 101: what are the Thai risk assessment tools? They are not described in detail so it is hard to know what they involve.

Introduction line 79: Can a brief description of what CAC involves and what it assesses be provided?

Methods line 116: I think the terminal nodes for the different strategies in the decision trees should be described. My understanding is that after screening, a patient can be on no statin, moderate potency statin, or high potency statin. This is clear in the figure but not described in the text. I think Table 1 could also be revised with a column added to show how these classifications relate to no/moderate/high potency statins. This will make it more clear for the reader.

Methods: Can more details be provided about who the cardiologists and health economists were who reviewed the model, and what the review process involved?

Table formatting looks off in my version. Will need to be revised to common font and layout in publication.

Table 2: What was cost of CAC testing? I don’t think this is shown in the costs section?

Table 2: Perhaps put non-medical costs in their own category so they are more obvious. Will also need to describe how these were gathered. Is it from a patient survey? There is very little detail on how these costs were obtained. Perhaps conducting a scenario analysis from the health system perspective would also be of interest.

Table 2: Relative risk compared to placebo – I think placebo is what it was compared to in the source trials, but here shouldn’t it be compared to “no statin”?

Table 2 and methods: reference 31 used for costs is from 2006. How were these costs inflated to present values? What year is the currency presented in? This is a requested item in the CHEERS list. Perhaps add a completed CHEERS checklist as an appendix.

Discussion Line 288: “Based on our analysis, approximately one-third of patients who followed the CAC screening strategy received moderate- or high-potency statins at the beginning, while the majority of patients in the current practice received moderate-potency statins (76%).“ As discussed in the methods feedback, would it be possible to show how many patients are on the different therapies after the decision tree portion of the model? It would make the differences in the treatment of the two groups more obvious and understandable.

Reviewers' comments:

Reviewer's Responses to Questions

**Comments to the Author**

Reviewer #2: (No Response)

2. Is the manuscript technically sound, and do the data support the conclusions?

Reviewer #2: Partly

3. Has the statistical analysis been performed appropriately and rigorously?

Reviewer #2: Yes

4. Have the authors made all data underlying the findings in their manuscript fully available?

Reviewer #2: Yes

5. Is the manuscript presented in an intelligible fashion and written in standard English?

Reviewer #2: No

Reviewer #2: There are numerous instances of incorrect grammar, it would be an enormous effort to correct them all as a reviewer. I note that the journal does not copyedit submissions but this should be done before proceeding with publication.

Here are more substantive comments:

Abstract: Should read “$160,00 BAHT per QALY” not QALYs

The statement: “Probabilistic, and additional one-way sensitivity analyses were performed to account for the model’s important assumption and robustness” is not clear. Suggest revising to say it is assessing robustness and testing the impact of assumptions.

Suggest rounding all monetary figures to the nearest whole value (i.e. remove decimals) throughout text.

Introduction line 69 and Methods line 101: what are the Thai risk assessment tools? They are not described in detail so it is hard to know what they involve.

Introduction line 79: Can a brief description of what CAC involves and what it assesses be provided?

Methods line 116: I think the terminal nodes for the different strategies in the decision trees should be described. My understanding is that after screening, a patient can be on no statin, moderate potency statin, or high potency statin. This is clear in the figure but not described in the text. I think Table 1 could also be revised with a column added to show how these classifications relate to no/moderate/high potency statins. This will make it more clear for the reader.

Methods: Can more details be provided about who the cardiologists and health economists were who reviewed the model, and what the review process involved?

Table formatting looks off in my version. Will need to be revised to common font and layout in publication.

Table 2: What was cost of CAC testing? I don’t think this is shown in the costs section?

Table 2: Perhaps put non-medical costs in their own category so they are more obvious. Will also need to describe how these were gathered. Is it from a patient survey? There is very little detail on how these costs were obtained. Perhaps conducting a scenario analysis from the health system perspective would also be of interest.

Table 2: Relative risk compared to placebo – I think placebo is what it was compared to in the source trials, but here shouldn’t it be compared to “no statin”?

Table 2 and methods: reference 31 used for costs is from 2006. How were these costs inflated to present values? What year is the currency presented in? This is a requested item in the CHEERS list. Perhaps add a completed CHEERS checklist as an appendix.

Discussion Line 288: “Based on our analysis, approximately one-third of patients who followed the CAC screening strategy received moderate- or high-potency statins at the beginning, while the majority of patients in the current practice received moderate-potency statins (76%).“ As discussed in the methods feedback, would it be possible to show how many patients are on the different therapies after the decision tree portion of the model? It would make the differences in the treatment of the two groups more obvious and understandable.

**Do you want your identity to be public for this peer review?** For information about this choice, including consent withdrawal, please see our Privacy Policy

Reviewer #2: No

---

## [Author Response · Author response to Decision Letter 2]

19 Mar 2025

Reviewers comment on the manuscript #2

Entitles: “Cost-utility analysis of Coronary Artery Calcium screening to guide statin prescription among intermediate risk patient in Thailand”

Dear Editor and reviewers,

We would like to thank you for your valuable reviews and comments. It is our great pleasure to have an opportunity to revise our manuscript. We have revised and modified our manuscript with some additional information (track change) as suggested by reviewers’ comment. We hope that our revisions will improve the quality of the manuscript and give a clearer vision of research methodology to meet qualification for publication in PLOS ONE. Please inform us if further information or clarification needs to be addressed. We are looking forward to your reviews and would be extremely grateful to make our response.

Reviewer #2

There are numerous instances of incorrect grammar, it would be an enormous effort to correct them all as a reviewer. I note that the journal does not copyedit submissions but this should be done before proceeding with publication.

Response: Thank you for your comments. We have carefully reviewed the manuscript and corrected all grammatical errors as suggested.

1. Abstract: Should read “$160,00 BAHT per QALY” not QALYs

Answer: Thank you for your suggestions. We have made a necessary edit throughout the manuscript.

2. The statement: “Probabilistic, and additional one-way sensitivity analyses were performed to account for the model’s important assumption and robustness” is not clear. Suggest revising to say it is assessing robustness and testing the impact of assumptions.

Answer: Thank you for your suggestions. We have edited according to your suggestion in line 46-49 as follows: “Probabilistic and one-way sensitivity analyses were conducted to assess the robustness of the model and evaluate how variations in key assumptions impact the results. These analyses help determine the reliability of the findings by exploring the extent to which changes in input parameters influence the overall conclusions.”

3. Suggest rounding all monetary figures to the nearest whole value (i.e. remove decimals) throughout text.

Answer: Thank you for your suggestions. We have made a necessary edit throughout the manuscript.

4. Introduction line 69 and Methods line 101: what are the Thai risk assessment tools? They are not described in detail so it is hard to know what they involve.

Answer: We appreciated your feedback. Thai risk assessment tool is called Thai CV risk tools [1]. The tool predicted 10-year risk of having cardiovascular event as percentage.

For more clarify, we have added more detail of the tools in line 68-72 as follows: “In subclinical individuals, primary prevention of ASCVD is based on the predicted 10-year risk of a CVD event. In Thailand, patients are assessed using Thai cardiovascular (CV) risk tools, which estimate their 10-year risk as a percentage [7]. Statin therapy is not recommended for individuals at low risk (<10% 10-year risk), while it is recommended for those at high ASCVD risk (>20% 10-year risk) [8].”

Reference

1. Vathesatogkit, P., Woodward, M., Tanomsup, S., Ratanachaiwong, W., Vanavanan, S., Yamwong, S., & Sritara, P. (2012). Cohort profile: the electricity generating authority of Thailand study. International journal of epidemiology, 41(2), 359–365. https://doi.org/10.1093/ije/dyq218

5. Introduction line 79: Can a brief description of what CAC involves and what it assesses be provided?

Answer: Thank you for your suggestions. We have added the short description of CAC in line 80-84 as follows: “The Coronary Artery Calcium (CAC) score, introduced in the late 1990s, is a non-invasive imaging technique used to assess coronary artery calcification [14]. It quantifies calcified atherosclerotic plaques within the coronary arteries and has been strongly associated with CVD events [15, 16]. Extensive evidence strongly supports the potential of Coronary Artery Calcium (CAC) scores as an accurate tool for cardiovascular risk stratification [17, 18].”

6. Methods line 116: I think the terminal nodes for the different strategies in the decision trees should be described. My understanding is that after screening, a patient can be on no statin, moderate potency statin, or high potency statin. This is clear in the figure but not described in the text. I think Table 1 could also be revised with a column added to show how these classifications relate to no/moderate/high potency statins. This will make it more clear for the reader.

Answer: Thank you for your suggestions. To ensure greater clarity and comprehensibility, we have added the corresponding treatment column for each terminal node in decision tree within Table 1, as presented below.

Table 1. Selected input parameters to the decision tree model and corresponding treatment strategy

Input parameter: probabilities Base-case value

mean (SE) Receiving treatment

Current practice strategy

Incidence of initial LDL-C ≥ 190 mg/dL 0.018 (0.013) High potency statin

Incidence of initial LDL-C 70-189 mg/dL 0.759 (0.04) Moderate potency statin

Incidence of initial LDL-C <70 mg/dL 0.223 (0.04) No statin

CAC screening strategy

Incidence of initial CAC ≥ 100 0.36 (0.013) High potency statin

Incidence of initial CAC 1-99 0.39 (0.013) Moderate potency statin

Incidence of initial CAC = 0 0.26 (0.012) No statin

*The input data were sourced from the hospital database from 2012 to 2022. A beta distribution was selected for these input parameters in the probabilistic sensitivity analyses. Abbreviation: CAC, coronary calcium score; LDL-C, low-density lipoprotein cholesterol; SE, standard error

7. Discussion Line 288: “Based on our analysis, approximately one-third of patients who followed the CAC screening strategy received moderate- or high-potency statins at the beginning, while the majority of patients in the current practice received moderate-potency statins (76%).“ As discussed in the methods feedback, would it be possible to show how many patients are on the different therapies after the decision tree portion of the model? It would make the differences in the treatment of the two groups more obvious and understandable.

Answer: Thank you for your suggestions. We have added the corresponding treatment column for each terminal node in decision tree within Table 1, as presented below.

Table 1. Selected input parameters to the decision tree model and corresponding treatment strategy

Input parameter: probabilities Base-case value

mean (SE) Receiving treatment

Current practice strategy

Incidence of initial LDL-C ≥ 190 mg/dL 0.018 (0.013) High potency statin

Incidence of initial LDL-C 70-189 mg/dL 0.759 (0.04) Moderate potency statin

Incidence of initial LDL-C <70 mg/dL 0.223 (0.04) No statin

CAC screening strategy

Incidence of initial CAC ≥ 100 0.36 (0.013) High potency statin

Incidence of initial CAC 1-99 0.39 (0.013) Moderate potency statin

Incidence of initial CAC = 0 0.26 (0.012) No statin

*The input data were sourced from the hospital database from 2012 to 2022. A beta distribution was selected for these input parameters in the probabilistic sensitivity analyses. Abbreviation: CAC, coronary calcium score; LDL-C, low-density lipoprotein cholesterol; SE, standard error

8. Methods: Can more details be provided about who the cardiologists and health economists were who reviewed the model, and what the review process involved?

Answer: The conceptualisation and initial drafting of the model structure were carried out with the valuable contributions of cardiologist Pannipa Suwannasom. The study’s model was then reviewed and validated by external cardiologists, including Songsak Kiatchoosakun, Suphot Srimahachota, Arintaya Phrommintikul, and Yotsawee Chotechuang. Furthermore, the economic aspects of the model were thoroughly reviewed by Surasak Saokaew. The process involved multiple meetings, beginning with designing the model structure and validating the data used for input parameters. Follow-up meetings were then conducted to review and discuss both the preliminary and final results.

We have acknowledged the external experts who assisted in this study in the Acknowledgements section, line 378 - 382, as follows: “This study was partially supported by the Faculty of Medicine, Chiang Mai University, the University of Phayao, and the Thailand Science Research and Innovation Fund. We extend our deep gratitude to cardiologists Songsak Kiatchoosakun, Suphot Srimahachota, Arintaya Phrommintikul, and Yotsawee Chotechuang for their valuable review and validation of the model.”

9. Table formatting looks off in my version. Will need to be revised to common font and layout in publication.

Answer: We appreciate your feedback. The font in all tables has been revised accordingly.

10. Table 2: What was cost of CAC testing? I don’t think this is shown in the costs section?

Answer: We appreciate your feedback and sincerely apologise for any confusion in our manuscript. The cost of CAC testing is provided in Table 2. However, we initially used the term "CT coronary calcium scan," which may not be familiar to experts outside the field. To enhance clarity and consistency, we have revised it to "CAC screening" throughout the manuscript.

11. Table 2: Perhaps put non-medical costs in their own category so they are more obvious. Will also need to describe how these were gathered. Is it from a patient survey? There is very little detail on how these costs were obtained. Perhaps conducting a scenario analysis from the health system perspective would also be of interest.

Answer: We appreciate your feedback and would like to address your concern regarding the non-medical costs associated with non-fatal MI. These costs were sourced from Anukoolsawat et al. (2006) [1], where interview records from 193 ACS patients were used to estimate direct non-medical expenses. The non-fatal stroke costs was based on data from Rattanavipapong et al. (2022) [2]. This study utilized retrospective patient data from a multicentre stroke unit, incorporating key parameters such as inpatient length of stay, number of follow-up visits, and travel distance to estimate non-medical costs. Furthermore, transportation, caregiver, food, and accommodation expenses incurred during treatment were referenced from Singhpoo K et al. (2009) [3]. To enhance transparency, we have added these details into the supplementary data on page 8-9, as outlined below.: “Details of Cost Parameters from Anukoolsawat et al. (2006) and Rattanavipapong et.al. (2022)

Our study derived costs for secondary treatment and prevention in populations who experienced non-fatal MI costs, fatal MI costs, and direct non-medical costs from Anukoolsawat et al. (2006) [9]. This study used data from the Thai Acute Coronary Syndrome (ACS) registry to estimate lifetime costs of ACS. Medical records of 330 ACS patients were used to calculate direct medical costs (pre-event and MI-related costs of non-fatal MI), while interview records of 193 ACS patients were used to estimate direct non-medical costs. For the costs of non-fatal stroke and recurrent non-fatal stroke (both direct and indirect), our study utilized data from Rattanavipapong et al. (2022) [8] due to its recency and comprehensive. This study utilized retrospective patient data from a multicentre stroke unit, incorporating key parameters such as inpatient length of stay, number of follow-up visits, and travel distance to estimate non-medical costs. Furthermore, transportation, caregiver, food, and accommodation expenses incurred during treatment were referenced from Singhpoo K et al. (2009) [11]. Detailed information on direct and non-medical costs and their components can be found in the supplementary materials of the original article.”

We performed sensitivity analysis using health care providers perspective. The incremental cost, QALYs, and ICER for CAC screening are presented in the table below. CAC screening remains cost-effective under Thailand’s willingness-to-pay threshold of 160,000 THB per QALY gained.

Scenario Incremental

Cost (THB) Incremental QALYs (years) ICER Decision*

Perspective

Healthcare provider perspective 24,762 0.62 40,020 CAC is cost-effective

Societal perspective (base-case) 10,091 0.62 16,308 CAC is cost-effective

We have added the relevant text addressing this issue in the Methods section, line 236-237, as follows: “The post-hoc sensitivity analysis using the healthcare provider’s perspective was performed to account for the impact of direct non-medical costs.”

And in result section, line 286-288 as follows: “The post-hoc sensitivity analysis using the healthcare provider’s perspective demonstrated that CAC screening remains a cost-effective strategy (Table 4).”

1. Anukoolsawat, Pongchai, Piyamitr Sritara and Yot Teerawattananon. “Costs of Lifetime Treatment of Acute Coronary Syndrome at Ramathibodi Hospital.” (2006).

2. Rattanavipapong W, Worakijthamrongchai T, Soboon B, et al. Economic evaluation of endovascular treatment for acute ischaemic stroke in Thailand. BMJ Open2022;12:e064403. doi:10.1136/bmjopen-2022-064403

3. Singhpoo K, Tiamkao S, Ariyanuchitkul S, Sangpongsanon S, Kamsa-ard S, Lekbunyasin O, Soommart Y. The expenditures of stroke outpatients at Srinagarind hospital. Srinagarind Medical Journal. 2009;24(1):54-9.

12. Table 2: Relative risk compared to placebo – I think placebo is what it was compared to in the source trials, but here shouldn’t it be compared to “no statin”?

Answer: Thank you for your suggestions. We have edited according to your suggestions.

13. Table 2 and methods: reference 31 used for costs is from 2006. How were these costs inflated to present values? What year is the currency presented in? This is a requested item in the CHEERS list. Perhaps add a completed CHEERS checklist as an appendix.

Answer: We appreciate your feedback and would like to address your concern. All Cost data was converted to 2024 values using the Thai consumer price index (CPI) [1] and are expressed in Thai Baht (THB) (approximately 36 THB = 1 US$ in 2024) [2]. To account for time preference in the model, future costs and QALYs beyond the first year were discounted by 3% annually. This information has been explicitly stated in method section, line 185-188 as follows: “Cost data were converted to 2024 values using the Thai consumer price index (CPI) [39] and presented in Thai Baht (THB) (approximately 36 THB = 1 US$ in 2024) [40]. Future costs and QALYs beyond the first year were discounted by 3% annually.” According to your suggestion, we have added the CHEERS checklist as a supplementary file.

Reference

1. Bureau of Trade & Economics Indices Ministry of Commerce. Consumer price index of Thailand year 2024 base year 2006, 2018, 2021, and 2023. Available: http://www.price.moc.go.th/price/cpi/index_new_all.asp [Accessed 8 January 2024].

2. International Monetary Fund. World Economic Outlook Database for October 2007. Available from: https://www.imf.org/external/np/fin/ert/GUI/Pages/Report.aspx [Accessed 10 May , 2024].

---

## [Decision Letter · Decision Letter 2]

24 Jul 2025

Dear Dr. Phinyo,

Thank you for submitting your manuscript to PLOS ONE. After careful consideration, we feel that it has merit but does not fully meet PLOS ONE’s publication criteria as it currently stands. Therefore, we invite you to submit a revised version of the manuscript that addresses the points raised during the review process.

We look forward to receiving your revised manuscript.

Kind regards,

Forgive Avorgbedor

Academic Editor

PLOS ONE

Journal Requirements:

Reviewers' comments:

Reviewer's Responses to Questions

**Comments to the Author**

Reviewer #2: All comments have been addressed

2. Is the manuscript technically sound, and do the data support the conclusions?

Reviewer #2: Yes

3. Has the statistical analysis been performed appropriately and rigorously?

Reviewer #2: Yes

4. Have the authors made all data underlying the findings in their manuscript fully available?

Reviewer #2: Yes

5. Is the manuscript presented in an intelligible fashion and written in standard English?

Reviewer #2: Yes

Reviewer #2: Congratulations on the hard work, this version is an improvement on the first one and reads very well. A few specific comments:

Line 50: has an extra zero been added to the incremental cost? Should be $10,000 not $100,000?

Line 177: What are the indirect costs based on?

Line 221: My jurisdiction’s guidelines also suggest a scenario analysis where the discount rate is set to zero (and one were it’s increased above the base case value). Could consider adding this but may not be expected in your jurisdiction.

**Do you want your identity to be public for this peer review?** For information about this choice, including consent withdrawal, please see our Privacy Policy

Reviewer #2: **Yes: ** Rebecca Hancock-Howard

---

## [Author Response · Author response to Decision Letter 3]

28 Jul 2025

This study was partially funded by the University of Phayao and Thailand Science Research and Innovation Fund (Fundamental Fund 2025, Grant No. 5017/2567).The funders had no role in study design, data collection and analysis, decision to publish, or preparation of the manuscript.

We have removed declarative statements, such as Data Availability, Competing/Conflict of Interest, Funding/Financial Disclosure, Consent to publish statements, etc. from the manuscript files.

Best,

Phichayut

---

## [Editor Report · Decision Letter 3]

1 Aug 2025

Cost-utility analysis of Coronary Artery Calcium screening to guide statin prescription among intermediate-risk patients in Thailand

PONE-D-24-31252R3

Dear Dr. Phichayut Phinyo,

We’re pleased to inform you that your manuscript has been judged scientifically suitable for publication and will be formally accepted for publication once it meets all outstanding technical requirements.

Kind regards,

Forgive Avorgbedor

Academic Editor

PLOS ONE
---

## [Editor Report · Acceptance letter]

PONE-D-24-31252R3

PLOS ONE

Dear Dr. Phinyo,

I'm pleased to inform you that your manuscript has been deemed suitable for publication in PLOS ONE. Congratulations! Your manuscript is now being handed over to our production team.

Kind regards,

on behalf of

Dr. Forgive Avorgbedor

Academic Editor

PLOS ONE